# The Relationship between Work, Health and Job Performance for a Sustainable Working Life: A Case Study on Older Manual Employees in an Italian Steel Factory

**DOI:** 10.3390/ijerph192114586

**Published:** 2022-11-07

**Authors:** Federica Previtali, Eleonora Picco, Andrea Gragnano, Massimo Miglioretti

**Affiliations:** 1Faculty of Social Sciences, Tampere University, 33100 Tampere, Finland; 2Gerontology Research Centre, Tampere University, 33100 Tampere, Finland; 3Bicocca Center for Applied Psychology–BiCApP, Department of Psychology, University of Milano-Bicocca, 20126 Milan, Italy

**Keywords:** sustainability, older workers, health, job performance, work ability

## Abstract

Background: Supporting and retaining older workers has become a strategic management goal for companies, considering the ageing of the workforce and the prolongation of working lives. The relationship between health and work is especially crucial for older workers with manual tasks, considering the impact of long-standing health impairments in older age. Although different studies investigated the relationship between work ability and job performance, few studies have analysed the impact of workers’ capability to balance between health and work demands, including managerial and organisational support (work-health balance). Considering health as a dynamic balance between work and health demands influenced by both individual and environmental factors, we assess the mediator role of work-health balance in the relation between work ability and job performance, both self-reported and assessed by the supervisor. Methods: The study utilises data from a case study of 156 manual workers, who were 50 years old or older and employed in a steel company in Italy. Data were collected inside the company as an organiational initiative to support age diversity. Results: The findings show that work-health balance partially mediates the relationship between work ability and self-rated job performance, while it does not mediate the relationship with job performance as rated by the supervisor. Supervisor-rated job performance is positively associated with work ability, while it decreases with the increasing perceived incompatibility between work and health. Conclusion: A perceived balance between health and work is a strategic factor in increasing manual older workers’ job performance. For older workers, not only the perceived capability to work is important but also the organisational health climate and supervisor’s support. More studies are needed to verify if managers overlook the importance of health climate and support, as strategic elements that can foster performance for older employees.

## 1. Introduction

The ageing of the global population is a well-known socio-demographic phenomenon of our century. The United Nations defined the years between 2021 and 2030 as the “Decade of Healthy Ageing” [1] to politically raise awareness about ageing, educate about longevity and support the creation of a society for all ages. Governments are increasingly aware of this demographic transformation and are starting to accommodate policies for a growing number of older persons. For example, in Italy, a wide national project (2019–2022) was funded by the Department of Family Policies (DFP) to develop multilevel and participatory coordination of active ageing policies [2].

In the workforce, ageing is a relevant social transformation due to the growing number of older employees. The ageing of the workforce is caused by the decrease in younger workers and the political decision to increase the retirement age over the last decades [3]. In the workplace, age-related human resources practices are progressively introduced to facilitate the prolongation of careers, create an age-friendly work environment and support healthy ageing at work [4,5].

In Italy, 23.37% of the national population is over 65 years old, while the EU27 average is 20.79% [6]. Nevertheless, in Italy, the ageing phenomenon has not been widely addressed [2,7] and employment of older workers (52–64) is still under the EU average. In Italy, 53.3% of workers between 55 and 64 are employed while the EU27 average is 60.8% [6]. In this European country, the economic sector of manufacturing employs the highest number of workers, and it is also ageing fast. According to Labour Force Survey data [8], the percentage of workers over 50 years old has increased by 63% in 10 years, from 900,000 workers in 2010 to 1,411,000 workers in 2019, and the majority of these workers are between 50 and 59 years old. Therefore, it is strategic to analyse ageing within this Italian sector. 

One aspect that deserves attention in relation to the ageing of the workforce is the relationship between ageing and health and the insurgences of long-standing diseases in older age [9,10,11]. An inability to balance health and work demands endangers workers’ capability to maintain a productive working life and age healthy [12]. Research has shown that prolonged fatigue, which includes severity, concentration problems, decreased motivation and levels of physical activity, is an antecedent of retirement intention, and fatigue may be a pathway to early retirement in older workers [13].

According to the International Classification of Functioning, Disability and Health (ICF) developed by the WHO [14], the presence or absence of disability depends on the interaction of individual and contextual factors. This framework highlights that health is linked to the assessment of an individual status of health (physical, psychological, social) and personal capabilities, as well as to the limitations (or support) that contextual factors provide [15,16]. Based on ICF as a theoretical framework and the development of the concept of work-health balance [10,15], the relationship between health and work can be investigated as a matter of balance, and not of static assessment. Accordingly, the pathway between maintaining a good work ability, which assesses the individual status of mental and physical abilities on the basis of work demands, and having a good job performance can be widened (mediated) by including environmental factors (work-health balance), such as supervisors’ support and a work climate attentive to health promotion [10,15,17].

These environmental factors that allow a balance between work and health are important for older manual workers. Ageing manual workers face a risk of lacking this balance because they are at the crossroads of possible decreasing work ability due to age [18] and the physical demand embedded in their work positions [19]. A study done in the Netherlands found that manual labour, namely materials handling, was one of the most important risk factors for moderate and long durations of sick leave [19]. Thus, ensuring a sustainable balance between work and health (work-health balance) is a key strategy for retaining manual older workers and preventing long-standing health diseases.

For companies, redeploying workers due to physical inability to work is an extremely costly measure, hence they have an interest in understanding how to sustain a good balance between health and work [4]. Companies and managers, especially direct supervisors, have an active role in ensuring a sustainable balance between work and health, both at the individual and organisational levels. Hence, assessing whether supervisors’ support and organisational health climate play a role in the relationship between work ability and job performance is critical to foster sustainable employability for older workers. 

Another critical management issue is to support older workers in maintaining satisfactory job performance. Job performance is one of the most important indicators for companies [20]. Although research has found that age has varied effects on job performance [21], older workers are often stereotypically considered less productive [22]. Research shows that stereotypes about age have an impact on the evaluation of the job performance of older manual workers due to their possible declining physical capabilities [22,23,24]. Due to the physical demands of manual work, a satisfactory performance is also critically linked to the workers’ work ability and capability to balance health and work demands. As such, job performance is a contested measurement and, especially for older workers, there is a need to understand its determinants and overcome stereotypical conceptualisation of age. 

In heavy and manual work positions, workers need to maintain high performance, despite the possible decrease in work ability due to ageing [9,25]. A population study demonstrated a significant negative association between age and work ability, with a drastic decrease for the population over 60 years old. However, there is great variability among individuals [9]. Good work ability is associated with a high quality of work, high productivity, and work engagement [26]. Despite the extensive research on work ability, Ilmarinen [25] stated that more research is needed to understand whether a poor balance between health and work is merely due to the decline of human resources due to ageing or to problems in work and organisation management. In this regard, work-health balance is a useful addition to the conceptualisation and assessment of the balance between health and work [10]. Work-health balance has proven to be associated with a series of job outcomes, such as affective job satisfaction [10], work engagement [17], and low levels of presenteeism and emotional exhaustion [11,17,25].

This paper aims to show that the perceived possibility for older workers to actively and effectively balance their work and health demands (work-health balance) mediates the effect of work ability on job performance, both self-assessed and assessed by the direct supervisor. Analysing this relationship is especially important to contribute to the retention of older workers in an ageing sector, such as manufacturing in Italy, and to develop interventions that support a sustainable extended working life, considering the raising of the retirement age. By considering health as a dynamic element instead of a static status [15], the ability to actively balance work and health demands can be considered to have a mediating role between work ability and job performance. Work ability is the ability to stay at work and manage work demands, considering one’s health status. Work-health balance adds to work ability contextual factors, such as flexibility, support and a health organisational climate that are fundamental aspects of a conceptualisation of health as derived from individual and contextual factors [14]. 

### 1.1. Health Promotion at Work: From Work Ability to Work-Health Balance

#### 1.1.1. The Relation between Work Ability and Job Performance

The link between health and work is strategic for the age-related management of workers and a key element to support sustainable labour participation [4]. Work ability (WA) is one of the first measures developed to assess the relationship between workers’ health and job demands. It measures a worker’s ability to meet the requirements of work, given their physical and mental state and health (e.g., WAI, work ability index [27]). The index was introduced in the 1980s in Finland with the aim of providing a new assessment tool for occupational health in order to support the prolongment of working life and increasing the older labour force [25]. Despite contradicting studies [25], on a general level, work ability is found to decline with ageing [25]. A decreased work ability is linked to reduced productivity at work, increased sick absences and early retirement. 

Therefore, we hypothesise:

**Hypothesis** **1 (H1)**.
*Work ability has a positive relationship with self-rated job performance.*


**Hypothesis** **2 (H2)**.
*Work ability has a positive relation with supervisor-evaluated job performance.*


#### 1.1.2. The Relation between Work Ability, Work-Health Balance and Job Performance

Due to the ageing of the workforce, there is an increasing interest in exploring the dynamics that link decreasing physical and mental capabilities with age and organisational outcomes. This interest is also caused by increasing academic awareness about health at work, as a complex phenomenon not only calculated on health status but involving a series of personal, organisational, social, and relational factors [15,16]. A holistic view of health promotion was introduced in the latest development of work ability (e.g., work ability house [25]). However, the index used in most academic studies and occupational interventions still assesses only health and mental status [9,11,17,26,28,29,30]. 

Alongside work ability, another measure was introduced in Italy to assess the health and work status among ageing workers and workers with long-standing health conditions: the work-health balance (WHB) [10]. Therein, health is conceptualised not as a status but in a dynamic manner, where every worker is an active actor in maintaining a good balance [15]. The balancing between health and work is not just a mere balancing of status but an ongoing negotiation and modelling of the borders between health and work domains (as conceptualised by Clark [16]), which have reciprocal influence. Hence, work-health balance is defined as the situation where workers believe that they can effectively balance their health and work demands [10]. WHB includes on one side an assessment of perceived incompatibility between health and work and on the other side two dimensions measuring the perception of helpfulness of the working environment, the workplace health climate and the support received from supervisors on health matters [10].

The work-health balance concept is rather new, therefore few studies explored it [10,11,17,31]. Nevertheless, all studies so far have found that WHB has a positive relation with diverse organisational outcomes. For example, WHB had a negative relation with presenteeism, and a positive one with performance and perceived health [10]. Gragnano et al. [17] showed that work-health balance was related to job satisfaction and this relation was significantly greater among workers with lower work ability. 

Therefore, we hypothesise:

**Hypothesis** **3 (H3)**.
*Work ability has a positive relation with work-health balance.*


**Hypothesis** **4 (H4)**.
*Work-health balance has a positive relation with self-evaluated job performance.*


**Hypothesis** **5 (H5)**.
*Work-health balance has a positive relation with job performance as evaluated by supervisors.*


#### 1.1.3. Job Performance: A Discussed Measurement

Job performance is critical for companies as well as for organisational psychologists because it is the closest way to assess organisational efficiency through scales and surveys handled by workers [32]. Nevertheless, the fact that job performance is mainly evaluated through self-reported variables has been the focus of criticism, because it can be subject to desirability bias and partial evaluation [33,34]. Reports of supervisors’ assessments are considered to be a more objective evaluation of workers’ performance [35]. However, recently Kock [35] showed that anonymous self-reported job performance instruments are more adequate than official scores received from immediate supervisors from a measurement instrument quality perspective. He argued that supervisors’ evaluations might be less consistent because they are not anonymous. Therefore, supervisors may overestimate performance to avoid conflict or be a poorer evaluator due to biases toward certain employee groups.

In the framework of sustainable employability, supervisors have not only the duty of assessing performance but also to actively support workers in maintaining their productivity [30,36]. Inside organisations, supervisors have the role of implementing workplace policies and endorsing organisational culture. In the context of supporting a good balance between work and health, managers are at the frontline of allowing workers to take care of their health, for example, by showing support, allowing sick leaves, flexibility, and shaping employees’ tasks [10,11].

Amidst the discussion about performance assessment measurements, a congruence between supervisors’ and employees’ assessment of performance is increasingly important to ensure the sustainable growth of working environments [30,36]. In fact, within the research on sustainable employability, a necessity is to align the expectation between employer and employee to create a work environment that allows the maximum utilisation of each ability and sustainably prolongs working life [15,36,37]. A misalignment between managers’ and workers’ assessments might be considered a danger to the construction of sustainable working life and reveal different dynamics of influence for employees and managers.

In our analysis, we use two measurements of job performance: a self-report measurement of in-role behaviours and a report of supervisors’ annual scores. The in-role behaviour performance is studied to be the closest to the supervisor’s evaluation of the subordinate’s performance [34]. We investigate the mediating role of work-health balance between work ability and job performance. WHB broadens previous concepts as work ability and includes a personal assessment of what the organisation put in place to support workers’ health, overcoming the mere idea of being “able” and giving importance to climate, policies, and support. Previous research has investigated WHB as a mediator. Previtali et al. [31] applied work-health balance to assess organisational outcomes among ageing employees in Italy and they found that WHB was a mediator between job demands and exhaustion, and between job resources and engagement. Figueredo et al. [11] found that WHB partially mediated the effect of work ability on job satisfaction. Due to the nature of WHB (including both individual and external factors of balance between health) and the theoretical ICF framework understanding health as based on both individual and contextual factors, we consider that WHB has a mediator role between work ability and possible organisational outcomes (job performance). While WA measures primarily a balance between job demands and the health status (mental and physical) of workers [19], WHB considers these capabilities, assesses them through one dimension (incompatibility between work and health) and adds to them the role of organisational factor (flexibility and support, organisational health climate).

In literature, there has been an emphasis on understanding mediators for job performance [21], especially for older workers. Moreover, recent analyses have called for more research on the internal dynamics that link work ability to organisational outcomes for older workers, to contribute to a holistic concept of health promotion [30]. We hypothesise that work-health balance functions as a mediator between work ability and job performance. Increased work ability will increase job performance through the ability to positively balance health and work demands also considering the role of a contextual factor in influencing the relationship between work ability and job performance.

Therefore, we hypothesise:

**Hypothesis** **6 (H6).**
*Work-health balance mediates the relation between work ability and self-rated job performance.*


**Hypothesis** **7 (H7).**
*Work-health balance mediates the relation between work ability and job performance rated by supervisors.*


## 2. Materials and Methods

The study was based on a steel production company in Italy and the presented survey was part of the company’s “Age Diversity” actions to assess and improve the age inclusiveness of the organisation. The research took place between January 2018 and April 2018. Once the survey was defined, the survey was distributed to the target population with the help of line managers in pen and paper format. The study is cross-sectional, and the sample is not randomised. The survey was distributed to employees aged 50 years or over at the time of the study, working in manual roles. All so-defined older workers with a manual job position were included and they were distributed in three different plants in Italy. To participate in the study, the participants had to read and sign the informed consent. The data were treated with respect, anonymity and confidentiality in the delivery, collection, analysis, and reporting process by the research team. The lead researcher was available at any time to answer questions about the survey or the management of the data. The study was conducted following the GDPR and the ethical guidelines of the University of Milano-Bicocca. The study was distributed to 300 production workers and 156 completed and returned the paper survey (52% response rate). 

### 2.1. Measures

The demographic information collected by the paper survey were gender, age, education, work occupation and marital status. All the scales included in the questionnaire are validated, except the job performance as rated by supervisors. 

Work-health balance (WHB) was measured by the WHB Questionnaire (WHBq) [10]. The tool includes 17 items and three subscales: Work-health incompatibility (WHI): a personal evaluation of the level at which work activities interfere with the management of one’s health. It comprises 6 items, such as “Your job is a hindrance to your health” (Cronbach’s alpha: 0.88). Health climate (HC): the perception of the attention paid by management to employees’ health. It comprises 5 items, for example, “Senior management acts decisively when health concerns emerge among employees” (Cronbach’s alpha: 0.91). External support (ES): the perception of the supervisor’s support regarding health needs. It comprises 6 items, for example, “Your supervisor allows you to arrive and depart from work when you want to for health reasons” (Cronbach’s alpha: 0.83). The total WHB score is calculated through a mathematical formula by subtracting WHI from the mean of HC and ES. The subscales include ratings on a five-point scale from 1 (completely disagree) to 5 (completely agree) for WHI and from 1 (never) to 5 (always) for HC and ES. A higher score on the final index means a better-perceived balance between health and work. 

Work ability was measured by the Work Ability Index (WAI) (Ilmarinen, 2007). The tool comprises seven items of testing: current work ability compared to lifetime best (0–10), work ability in relation to the demands of the job, number of current diseases diagnosed by a physician, estimated work impairment due to diseases, sick leave during the past 12 months, the personal prognosis of work ability 2 years from now and mental resources. The final WAI score ranges from 7 to 49, based on the sum of the individual items, and a higher score indicates higher work ability. 

The self-rated job performance (JPself) was measured on a 6-item questionnaire with a five-point Likert scale. The 6 items measure in-role behaviour and are one subcomponent of the job performance scale by Anderson and Williams (1991). The scale measures the worker’s compliance with their organisational role and includes items such as “I fulfil the responsibilities specified in my job descriptions” (Cronbach’s alpha: 0.85). The answers are given on a five-point rating scale from 1 (completely disagree) to 5 (strongly agree). 

The job performance evaluated by the supervisor (JPeval) was retrieved by the formal performance evaluation received by each worker by their supervisors. The workers were asked to report the score (from 1 to 5) that they received in their last formal performance evaluation. In the company where the study took place, the workers received a yearly performance evaluation by their direct supervisors and this was in the range from 1, the lowest, to 5, the highest. This single score influenced their annual bonus.

We used age, education, and work occupation as control variables. 

### 2.2. Data Analyses

First the descriptive and correlation analyses were carried out using the SPSS 26 software package. Next, a mediation analysis was performed to confirm the direct association between work ability and both job performance measures (Hypotheses 1 and 2) and work-health balance (Hypothesis 3), as well as between work-health balance and both job performances measures (Hypotheses 4 and 5) and the indirect effects of work-health balance on the relationship between work ability and self-rated and supervisors’ evaluated job performance (Hypotheses 6 and 7).

To determine whether work-health balance mediated the relationship between work ability and job performance (both measures, one at a time), two mediation analyses (model 4) were carried out using the Process macro for SPSS [38]. The bootstrapping technique with 5000 subsamples was used to estimate the confidence interval (95%). This macro provides estimates of the indirect effects of the mediator, along with standard errors (SE) and confidence intervals (CI). The technique used is bootstrapping, a non-parametric re-sampling procedure that does not impose an assumption of normality on the sample distribution. Indirect effects are considered statistically significant if the confidence intervals (CI of 95%) do not contain zero [38]. Age, education, and work occupation were included as control variables. Gender was not used as the sample is composed of 97.6% by men (as shown in Table 1). Age was added as a control variable to control for the possible influence of age on self-assessed and supervisors-assessed job performance. According to the model of Selection, Compensation and Optimisation [39], older workers might focus on tasks that are better suited for them, hence reporting higher self-assessed performance. While according to age stereotypes literature [40], supervisors might give lower performance ratings to older workers. Lastly, work occupation was added as a control variable because of the possible difference in manual labour among the positions. Moreover, different work occupations have different supervisors, so by controlling for work occupation we control for possible group variation among supervisors. 

## 3. Results

### 3.1. Descriptive and Correlation Analyses

First, descriptive analyses were conducted to show the features of the samples. As shown in Table 1, 96.7% are male, and the age varies, most of the sample (66.7%) was between 50 and 55 years old, 69.9% had a secondary school education and 55.1% worked in a production position.

Second, bivariate correlations were analysed to determine the relationships among the variables. Means, standard deviations and correlations for the JPself scale, the WHB index and the WAI and JPeval scores for the samples are presented in Table 2. Work ability was directly and positively associated with work-health balance (r = 0.49), self-rated job performance (r = 0.30) and job performance evaluated by the supervisor (r = 0.40). Work-health balance was directly and positively associated with self-evaluated job performance (r = 0.37) and job performance evaluated by supervisors (r = 0.22). Moreover, the two measurements of job performance were positively associated (r = 0.31). 

### 3.2. Mediation Analyses

As shown in Table 3, the mediation analysis showed that: 1. there was a direct association between work ability and self-rated job performance (β = 0.18, SE = 0.09, *p* < 0.05), hence H1 is confirmed; 2. there was a direct association between work ability and work-health balance (β = 0.50, SE = 0.08, *p* < 0.001), hence H3 is confirmed; 3. the mediator work-health balance has a direct association with the outcome self-rated job performance (β = 0.32, SE = 0.09, *p* < 0.01;), hence H4 is confirmed. In the analysis of the global model, the indirect effect of work ability on self-rated job performance was found to be significant, indicating a partial mediation effect (β = 0.16 SE = 0.06, 95% CI [0.05–0.29]). Therefore, H6 is confirmed, that work-health balance partially mediates the relation between work ability and self-rated job performance. There is a direct and positive association between self-assessed job performance and age. There is also an effect of work position. Workers in production have a higher self-assessed job performance compared to workers in logistics, with self-assessed job performance of workers in maintenance as a reference.

As shown in Table 4, we run a mediation analysis between work ability and job performance evaluated by the supervisor, with work-health balance as a mediator. Work ability had a positive relation with job evaluation (β = 0.53, SE = 0.08, *p* < 0.001), hence H2 is confirmed. Nevertheless, the work-health balance did not have a significant relationship with job performance evaluated by the supervisor (*p* = 0.14, SE = 0.9), thus H5 is not confirmed. Consequentially the meditation is not verified, hence H7 is also not confirmed (β = 0.07, SE= 0.05, CI [−0.01–0.19]).

## 4. Discussion

This study aimed at evaluating work-health balance as a mediator in the relationship between work ability and job performance as self-evaluated and evaluated by the supervisor in a sample of older manual workers in a steel factory in Italy. Our study confirmed that work-health balance functions as a partial mediator between work ability and self-rated job performance. However, one of our hypotheses was not confirmed and work-health balance could not be considered as a mediator between work ability and job performance as evaluated by supervisors [11]. 

This study contributes to the literature on prolonging working lives by focusing on the dynamics surrounding the balance between health and work demands for older manual workers. We found that work ability positively influences workers’ own evaluation of job performance, and this relationship is mediated by a positive balance between health and work demands (work-health balance) for older manual workers. This is in line with the theoretical understanding that balancing health and work demands is a dynamic equilibrium [15,16,17]. The pathway between work ability and job performance is also influenced and supported by the personal perception of this equilibrium, where contextual factors are strategic. As such, work-health balance is confirmed to be a useful tool to assess the capabilities of workers to manage their health and the support given by organisation and supervisors in doing so [11]. In practice, the measurement of work-health balance adds to the measurement of work ability index, which has a stronger focus on individual factors. These results are in line with previous studies that showed how work-health balance is a mediator between work ability and affective job satisfaction [11], or between job demands and emotional exhaustion [31].

This study investigates whether work-health balance has a mediation effect between work ability and two different measurements of job performance: self-rated and rated by supervisor. We did not hypothesise for a difference in the models (with self-assessed job performance and supervisor-assessed job performance), because in a sustainable organisation these two should be aligned [15,36]. Among our hypotheses, the assumption that work-health balance mediates the relation between work ability and job performance as evaluated by the supervisors was not confirmed. Nevertheless, work-health balance correlates with job performance as evaluated by supervisors (r = 0.40). 

We might hypothesise that, when the workers perceive an incompatibility between work and health demands, this may lead to a decreased job evaluation from supervisors. Moreover, in the mediation analysis, work ability has a stronger effect (linear regression) on job performance as evaluated by the supervisor than on job performance as self-evaluated. Results may suggest that managers are negatively influenced in their assessment of older manual workers’ performance by employees’ work ability and do not take into consideration the role of environmental factors in influencing workers’ performance. Managers are less subjected to more soft influences, such as organisational factors, in their evaluations of employees. This might be linked to the literature on age stereotypes and the idea that managers provide an evaluation based on an image of older workers where the health impairments (as assessed by work ability) have a strong influence [4,41,42,43,44]. Previous research has found that while managers’ attitudes toward soft skills (reliability and loyalty) have improved over time, their attitudes toward hard skills (such as physical stamina) have not improved [24]. Therefore, work ability is influential for supervisors’ job performance evaluation in positions that involve manual labour and physical activity.

The fact that work-health balance mediates only one model can also be interpreted as a demonstration that self-assessed job performance and supervisor-assessed job performance do not measure the same elements [37]. This might also be supported by the fact that the correlation between self-assessed job performance and supervisor evaluation is relatively low (r = 0.31, *p* < 0.01). 

These results shed new light on the mechanisms that boost manual workers’ productivity and contribute to the understanding that sustainable employability is a result of diverse factors, not only health status. The results show that improving organisational factors and climate can increase older workers’ self-assessment of performance. Self-assessed job performance is linked to increased job satisfaction and retention [33], and studies have argued that it is a more proper evaluation of productivity because less influenced by possible biases, compared to supervisors’ assessments [35]. A good health climate and external support are part of the organisational factors that can ensure sustainable employability and boost productivity for older workers [15,38]. The result shows that creating an organisational climate where health is perceived as central and workers’ well-being is prioritised, is a resource to support older workers’ in-role performance. Moreover, age is a significant predictor of self-assessed job performance, which is in line with the SOC (Selection, Optimisation and Compensation) theory [39]. Older workers tend to select the domains on which to focus, optimise their potential in these domains and, in this way, compensate for age-related losses (for a review see [45]). Therefore, older workers might select job-related tasks that allow them to optimise their performance and compensate for age-related losses, and in these ways, have a higher perception of their job performance. The SOC model theorises for an agentic role of older workers in their productivity; this agentic view is also in line with the theoretical basis of work-health balance. Work-health balance assumes that workers actively balance the incompatibility of work and health with the perceived support by supervisors and organisation, and their possibility to take care of their health at work [10,15,17].

On the contrary, job performance assessed by the supervisors is not significantly predicated by age, while it is by work ability. This is in line with the literature on age in the workplace [46,47,48,49], which shows that chronological age is not the same as functional age, which is measured by work ability. Moreover, this might also be in line with ageism literature, and the assumption that managers perceive all older workers as the same, regardless of their actual age [48]. This is an important result to support the idea that policies regarding age management should not be based on chronological age groups but instead start from a more holistic understanding and assessment of age-related dynamics in the workplace.

### 4.1. Methodological Limitations

The sample of the study belonged to only one economic sector and included only workers over 50 years old. It is also unknown whether the sample was representative of the steel industry at a regional or national level. The response rate was quite low (52%), due to the difficulty of handing out paper-based surveys. We can assume that a selection bias occurred and only workers motivated to support organisational initiatives responded. Moreover, in Italy, the manufacturing sector is highly politicised, and the project was supported by the labour union. However, the topic of health and the possibility for the employer to ask about health are perceived as highly delicate and breaching the privacy of workers. The participants were mostly men, hence the gender unbalance is a limitation to the generalisation of the results, although manufacturing and manual work are gendered (male) labour markets in Italy. The sample is rather limited and, as such, the conclusion needs to be interpreted in light of this limitation.

The study used a cross-sectional design and not a randomised sample, which limits the possibility of showing causality, and the relationship studies could be subject to reverse causation. Thus, any arguments for causality need to be further proved in longitudinal studies.

The study does not address and evaluate whether and which ergonomic measures were in place at the time of the survey. Ergonomic solutions and other health prevention measures might have an influence on the assessment of work ability and the health work climate and be especially important for workers with manual work positions. Nevertheless, the narrow focus allows us to investigate the utility of specific measures and management practices and contribute to the contextualisation of ageing dynamics within the workforce, to overcome the “one size fits all approach” of age management [49].

### 4.2. Implications for Practice and Future Research

This study underlines that for older manual workers the influence of work ability on job performance, as self-assessed, is mediated by work-health balance. On the contrary, the influence of work ability on job performance, as assessed by supervisors, is not mediated by work-health balance. Moreover, work ability and work-health balance have a positive relationship with both job performance measurements, but work ability is more highly correlated with supervisor-rated job performance, while work-health is more highly correlated with self-assessed job performance. These results suggest that the pathways to boost job performance as self-assessed or assessed by supervisors are different.

The first practical implication of the study is that the work ability index and work-health balance questionnaire are useful tools to be used in organisational health promotion activities. While work ability is a very used measurement [25,27,31], work-health balance is not yet introduced as a tool in interventions. This study adds to previous literature [10,11,17,31] that shows how work-health balance plays a role in sustaining workers’ well-being and productivity. Therefore, the work-health balance questionnaire can be introduced as an assessment tool to sustain and support health intervention, especially for older workers. Work-health balance broadens the assessment of work ability by investigating not only if a worker is “able”, but also what a company can do for them [15].

A second practical implication is that in economic sectors that are ageing fast, such as manufacturing companies in Italy, working on promoting a good balance between health and work can support older manual workers. In fact, if a company fails to retain and support older workers, it might face a shortage of labour force and many unmotivated workers. A possible approach aimed at boosting manual older workers’ performance is to increase the work-health climate of the organisation, hence having more attention towards health-related issues and promoting more policies and initiatives in relation to health promotion, besides safety. The study shows that it is necessary to foster a healthy climate and educate managers about the importance of showing attention to workers’ health needs.

Third, the different results of the two mediation models suggest that a more consistent and shared understanding of what job performance means and what job performance indicators measure might foster sustainable employment throughout the life course [50,51,52,53]. This leads to a problematisation at organisational level of what job performance indicators measure.

Taken together, these findings suggest some further courses of action for companies’ management. First, improving the health climate will support older workers’ productivity, as we have shown that work-health balance mediates work ability and self-assessed job performance. Second, training managers about the difference between chronological age and functional age will decrease age stereotypes in the workplace [52,53,54]. In fact, we have shown that chronological age does not influence supervisor-assessed job performance, but work ability, which can be considered functional age, does. Third, giving managers more resources to support workers’ health-work balance will support them in the promotion of well-being in the workplace [54,55].

The question raised by this study call for further research investigating the perspective of managers and employees on health climate and work-health balance measure. Longitudinal studies can be considered to investigate the relations explored here and show casualty. Research involving larger samples will confirm the described relations. This research has raised questions about the symmetry of attitudes and the meaning of health and performance between supervisors and employees, and this question can be investigated through qualitative studies. Further studies can involve a more gender-balance sample to investigate the dynamic of gender on health and work balance.

## 5. Conclusions

The health status of older workers has been a central point for prolonging working lives among researchers and occupational health practitioners. Health is more than status or the assessment of being able to work, considering physical or mental issues. The well-being of workers also entails the possibility for them to manage their health demands at work and engage in an active, sustainable and responsible balance among different domains, including work and health. This study shows that wide organisational support towards health management and attention from management to workers’ well-being are key resources to boost older workers’ performance.

Overall, this study is relevant to the literature on healthy ageing at work and the sustainability of prolonged working lives. It sheds a new perspective on what sustainability and work-health balance mean. Moreover, it shows that implementing solutions to sustain well-being and older workers’ health is not enough if there is not a coherent match between employees’ and managers’ understanding of well-being and performance.

## Figures and Tables

**Table 1 ijerph-19-14586-t001:** Descriptive statistics of the total sample (N = 156).

Variable	%	(N)
*Gender*		(156)
Male	96.7	(151)
*Age*		
50–55 years	66.7	(104)
55–60 years	31.4	(49)
60–65 years	1.9	(3)
*Education*		(156)
Elementary	3.2	(5)
Secondary school	69.9	(109)
High school	26.3	(41)
University	0.6	(1)
*Work occupation*		(156)
Production	55.1	(86)
Machine maintenance	16.7	(26)
Logistics	28.2	(44)
*Marital status*		(156)
Single	15.4	(24)
Married/living with a partner	74.4	(116)
Divorced	8.3	(13)
Widow	1.9	(3)

**Table 2 ijerph-19-14586-t002:** Means, standard deviations (SD) and correlations for the WHB index, JPself scale and the WAI and JPeval scores (N = 156).

Measure	Mean	SD	1	2	3	4	5	6	7
1. Age	-	-	1						
2. Education	-	-	0.17 *	1					
3. Work occupation	-	-	0.15 *	0.02	1				
4. Work Ability Index (WAI)	36.7	5.76	−0.06	0.00	−0.09	1			
5. Work-Health Balance (WHB) index	1.27	1.44	0.04	0.10	0.03	0.49 **	1		
6. Self-rated job performance (Jpself)	4.11	0.53	0.20 *	0.08	−0.07	0.30 **	0.37 **	1	
7. Job performance evaluated by the supervisor (Jpeval)	3.40	1.06	0.03	0.07	0.02	0.40 **	0.22 **	0.31 **	1

Note: * *p* < 0.05. ** *p* < 0.01.

**Table 3 ijerph-19-14586-t003:** Indirect and Direct Associations between Work Ability, Self-Rated Job Performance (JPself) and Work-Health Balance.

	Work-Health Balance			JPself		
Variable	β	SE	*p*	β	SE	*p*
Age	0.15	0.16	ns	0.56 **	−0.16	0.00
Education	0.16	0.16	ns	0.02	0.16	ns
Machine maintenance-logistic	0.00	0.24	ns	0.12	0.24	ns
Production-logistic	−0.27	0.19	ns	0.38 *	0.19	0.05
Work ability	0.50 **	0.08	<0.001	0.18 *	0.09	0.05
Work-health balance				0.32 **	0.09	0.00
Model summary		R^2^ = 0.27 **			R^2^ = 0.29 **	
**Indirect effect of work ability on self-JP**	**Effect**	**Boot SE**	**Boot 95% CI**			
	0.16	0.06	[−0.05, 0.29]			

Note: * *p* < 0.05. ** *p* < 0.01. ns = non-significant. The remaining dummies for the covariate work occupation were included in a second mediation analysis, but the variable machine maintenance (in comparison with production) and logistic (in comparison with production) were not statistically significant. We decided to include only the variable production (in comparison with logistics) and machine maintenance (in comparison with logistics).

**Table 4 ijerph-19-14586-t004:** Indirect and Direct Associations between Work Ability, Job Performance Evaluated by Supervisor (JPeval) and Work-Health Balance.

	Work-Health Balance			JPeval		
Variable	β	SE	*p*	β	SE	*p*
Age	0.14	0.16	ns	0.03	−0.16	ns
Education	0.14	0.16	ns	0.06	0.17	ns
Machine maintenance-logistic	0.09	0.25	ns	−0.28	0.25	ns
Production-logistic	−0.25	0.19	ns	−0.21	0.19	ns
Work ability	0.53 **	0.08	<0.001	0.37 **	0.09	0.00
Work-health balance				0.14	0.09	ns
Model summary		R2 = 0.29 **			R2 = 0.22 **	
**Indirect effect of work ability on Jpeval**	**Effect**	**Boot SE**	**Boot 95% CI**			
	0.07	0.05	[−0.01, 0.19]			

Note: ** *p* < 0.01. ns = non-significant. The remaining dummies for the covariate work occupation were included in a second mediation analysis, but the variable machine maintenance (in comparison with production) and logistic (in comparison with production) were not statistically significant. We decided to include only the variable production (in comparison with logistics) and machine maintenance (in comparison with logistics).

## Data Availability

The data presented in this study are available on request from the corresponding author. The data are not publicly available due to informed consent statements.

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
