# Peer review of "The Relationship between Work, Health and Job Performance for a Sustainable Working Life: A Case Study on Older Manual Employees in an Italian Steel Factory"

_ijerph, 2022, doi:10.3390/ijerph192114586_

Round 1
Reviewer 1 Report
Excellent paper with important findings. My only suggestion is that some mention of the role heavy work plays in producing health limitations be made in the background, and the absence of inquiry into ergonomic or other prevention measures be made in the limitations section.
Author Response
"Please see the attachment."

Reviewer 2 Report
Line 24: “Supervisor-rated job performance is associated with work ability and perceived incompatibility between work and health.” Please specify the direction of associations.
Line 53 “Although research has shown that age has no clear and unambiguous effect on job performance” There’s risk of overstatement. It would be safer to say age has been found to have varied effects on job performance. (see Ng, T. W., & Feldman, D. C. (2008). The relationship of age to ten dimensions of job performance. Journal of applied psychology, 93(2), 392.).
Line 61 repeated “is strategic”
In line 72, consider adding the gap in literature mentioned above, especially reference number 13 and 14, which should be described in more details.
The correlation between self-rated job performance and supervisor-rated job performance seems quite low (.31), any explanation? The difference mediating results between the two needs to be further investigated and discussed.
The significance of this study’s major new finding (mediation effect) needs to be further addressed in introduction (rationales needs to be provided besides “not studied before”) and discussed in discussion (policy implication, organizational support, etc.).
My major concern is the main hypothesis of the study. The study hypothesized work-health balance to be a mediator of between work ability and job performance. Isn’t it makes more sense to hypothesize work-health balance as a moderator between the work ability and job performance?
Author Response
"Please see the attachment."

Reviewer 3 Report
This study has focused on an important phenomenon, aging population and work-life balance impacts on them. It has provided a coherent story with an appropriate research design. The findings are relevant to business/companies and could be extended to other countries/regions.
It will be helpful if the authors could elaborate a bit more based on existing literature, especially strengthening the theoretical foundations of those hypotheses. In addition, what could be the reason that H5 was not supported? Since the sample is from a manufacturing company and majority of the participants are male, this could be a limitation. It will be interesting to further explore those hypotheses in gender balanced companies.
Author Response
"Please see the attachment."

Reviewer 4 Report
This study investigates the mediation effect of work-health balance on the relationship between work ability and job performance within an older workforce in Italy. The manuscript is well written and interesting and in my opinion fits well into the scope of the journal. However, this study has more limitations than the authors admit. For example, the cross-sectional nature or the lack of control for confounders. The latter might be corrected by the authors, the other limitations should be acknowledged. Furthermore, I have some more minor comments.
Introduction
Line 61: typing error
Line 65-67: This sentence (especially the last part of the sentence) is not easy to understand. Please consider to rephrase
Line 67-69: Please consider to rephrase „to be made redundant“
Line 74: I do not feel that the definition of work ability is always correct in the manuscript. Work ability is a more complex than described here. For example in line 74 it is described as „the capability to work due to health conditions“. However, work ability is also dependant on job demands and mental resources (see for example: Ilmarinen et al. 2005. New dimensions of work ability. International Congress Series. 1280:3-7). Please reflect. See also lines 84-85. However, lines 86-87 correctly define work ability.
Line 88: I feel that an explanation is needed about what it meant by „to answer to the ageing of the workforce“.
Line 89: Previous research is not entirely clear about the relationship between work ability and age, which the authors have also accounted for in line 95. I therefore recommend to give a more careful statement here about this relationship.
Line 90: More references are needed here.
Lines 83 – 93: Since this paragraph is about the relationship between work ability and performance, the authors might consider to add another heading to this section. For example „The relationship between work ability and job performance“.
Lines 110-126: This section lacks references.
Line 142: What is meant by „reported measures“?
Line 148, Line 155: Comma wrong
Line 156-168: Please consider to rephrase. The sentence is hard to understand.
Line 210: Please consider rephrasing. For example: „Work-health balance comprised three factors: Work health incompatibility (WHI), Health climate (HC) and External support (ES).“
Methods
Within the method section some important information is missing:
· When did the survey took place?
· Have all older workers been asked to participate or only a subsample? How was this subsample selected?
· Some information about the operationalization of the work-health balance questionnaire is missing. For example, what does 1-5 of the Likert scale mean (e.g. „totally agree“ to „totally disagree“). Did you calculate a composite score over WHI, HC and ES and how ( e.g. mean or sum score)? What do higher and lower scores of this final scale mean? The same applies to self-rated job performance.
Can you estimate whether a selection bias occurred? Who were the older workers, who did not respond? Is it possible to compare non-responders with responders, for example regarding age, sex, supervisor-evaluated performance?
Please consider to control analyses for relevant confounders (for example: age, gender, physical activity, education, occupation, etc.).
Categorizing a continuous measure leads to a significant loss of information. Please consider to use the WAI as a measure from 7-49 or provide an explanation on why work ability was categorized. Furthermore, within the correlation table, the mean and sd of the WAI is given. However, if you really want to use the WAI as a categorized variable, please consider to also give the distribution of the four categories.
Please state whether the measures have been validated.
Results
Line 269: Typo „sample“ instead of „samples“
Within the text, the regression coefficient is named ß. In the table it is named B. I therefore recommend to use either ß oder B dependant on whether the standardized or unstandardized regression coefficient was used. Furthermore, the regression coefficient and the standard error of the relationship of work-health balance and self-rated job performance differs between the text and the table.
Within the table notes, the abbreviation of work-health balance is explained, but this abbreviation isn’t used within the table. Please correct.
Discussion
Line 311: The two sentences need to be separated by a comma, not a dot. However, the sentence might be too long then and therefore the authors might consider to rephrase.
Line 312: Why is it confirmed for a variety of organisational outcomes and why is there a reference given when only one outcome (self-evaluated performance) was confirmed in the own study? Please consider to rephrase.
Line 321: There is a word missing: „it“
Line 328-329: Even though two forms of job performance were used as a outcome, they were not compared within the study. I therefore feel that this sentence is not correct.
Line 341-343: I do not understand the meaning of this sentence. Please consider to rephrase.
The limitation section is very short and there are surely more limitations of the study that could have been added (for example: common method bias, cross-sectional study, reverse causation, missing control for confounders, small sample size). Please consider to expand this section.
Lines 355-358, lines 362-364: I do not feel that these sentences really reflect what the study was able to show. It only used one measure of work-health balance (not „different measures“) and also did not explore whether a misalignment between employees‘ and supervisors‘ expectations is present or wether it is associated with sustainability.
Lines 365-370: Please give some references for those statements.
Author Response
"Please see the attachment."
